# Molecular and Structural Effects of Percutaneous Interventions in Chronic Achilles Tendinopathy

**DOI:** 10.3390/ijms21197000

**Published:** 2020-09-23

**Authors:** Christelle Darrieutort-Laffite, Louis J. Soslowsky, Benoit Le Goff

**Affiliations:** 1Rheumatology Department, Nantes University Hospital, 44000 Nantes, France; benoit.legoff@chu-nantes.fr; 2INSERM UMR1238, Bone Sarcoma and Remodeling of Calcified Tissue, Nantes University, 44000 Nantes, France; 3McKay Orthopaedic Research Laboratory, University of Pennsylvania, Philadelphia, PA 19401-6081, USA; soslowsk@upenn.edu

**Keywords:** Achilles tendinopathy, ultrasound-guided injections, platelet-rich plasma, stem cells, dry needling

## Abstract

Achilles tendinopathy (AT) is a common problem, especially in people of working age, as well as in the elderly. Although the pathogenesis of tendinopathy is better known, therapeutic management of AT remains challenging. Various percutaneous treatments have been applied to tendon lesions: e.g., injectable treatments, platelet-rich plasma (PRP), corticosteroids, stem cells, MMP inhibitors, and anti-angiogenic agents), as well as percutaneous procedures without any injection (percutaneous soft tissue release and dry needling). In this review, we will describe and comment on data about the molecular and structural effects of these treatments obtained in vitro and in vivo and report their efficacy in clinical trials. Local treatments have some impact on neovascularization, inflammation or tissue remodeling in animal models, but evidence from clinical trials remains too weak to establish an accurate management plan, and further studies will be necessary to evaluate their value.

## 1. Introduction

Tendon pathologies are a common problem, especially in athletes and in people of working age, as well as in elderly patients. Rotator cuff tendinopathies and Achilles tendinopathies (AT) are the most frequent locations. For AT, the incidence rate is about 2.35 per 1000 persons in the adult population (21–60 years), and AT is related to the practice of sport in 1/3 of the cases [1]. Pain is the main reported symptom; it is often chronic and has an impact on the work and daily life of patients. Today, the management of musculoskeletal pathologies represents a major societal burden [2]. Among the etiological factors, intrinsic and extrinsic factors are highlighted. Intrinsic factors include architectural disorders of the foot and more general factors such as age, use of corticosteroids, obesity or intake of fluoroquinolones [3]. The main extrinsic factor is mechanical overload responsible for micro-traumatic lesions.

In response to overuse, the tendon tissue increases its metabolic activity: tenocytes multiply and upregulate the extracellular matrix (ECM) turnover. Under mechanical stress, there is an increased expression of matrix metalloproteinases (MMPs), responsible for collagen degradation and leading to an accelerated turn-over of the tendon matrix. During tendinopathies, the pathological region is distinct from normal tendon with both matrix and cellular changes: increased cellularity, cell rounding, disorganized collagen matrix, and increased infiltration of blood vessels [4]. Biochemical and molecular changes are also observed [4]: in the matrix, expression of types I and III collagen is increased as well as glycosaminoglycan, aggrecan, and biglycan. Type III collagen is produced in the initial phase of tendon damage to provide a rapid ‘patch’ but has inferior biomechanical strength and needs to be gradually replaced by type I collagen to obtain a more resistant tissue. In addition, there is an imbalance in the equilibrium between the activity of MMPs and tissue inhibitors of metalloproteinase (TIMPs) [5,6]. Finally, the role of inflammation and that of the immune system remain unclear in tendon healing [7]. Some molecular evidence suggests that key inflammatory interactions occur in the earlier (acute and subacute) stages of repetitive tendon micro trauma [8]. Inflammation-related genes (*MIF, IL-18, IL-6,* and *TNF*) are up-regulated when signs of tendinopathy are moderate [9], and an inflammatory cellular infiltrate composed of macrophages and mast cells has been observed [10,11,12].

To determine the mechanisms of tendinopathy, animal models have been developed. Rats represent a good species as an animal model of tendinosis. The size of their tendons allows surgical or percutaneous procedures to be performed and adequate amounts of tissue to be collected for laboratory analysis [13]. In addition, rats can be readily trained and submitted to repetitive exercise. Animals are subjected to this mechanical overload through the use of forced treadmill running. However, it has variable success in creating overuse tendon pathologies. Treadmill running successfully induces tendinosis in the rat supraspinatus tendon [14] but it shows variable success in producing the pathological features in the Achilles tendon [15,16]. In addition, it can be difficult to force all rodents to run sufficiently fast or intensely to reach the level necessary for overuse, and the time required for the production of tendinopathy is long. Considering that, researchers have investigated the role of intrinsic factors in the development of tendinosis in animal models. This typically involves intratendinous injection of chemical compounds, such as collagenase, prostaglandin E1 (PGE1) or prostaglandin E2 (PGE2) [17]. Collagenase injection induces tendon degeneration through the breakdown of collagen. However, intratendinous injection of collagenase results in an acute and intense inflammatory reaction followed by spontaneous tendon reparation. This does not replicate the degenerative pathology (tendinosis) observed in humans. Because it is fast and easy to set up, collagenase-induced tendinopathy is the most widely used model in treatment studies despite this limitation. In animal models of tendinopathy, the main outcomes are the assessment of the tendon structure and its mechanical properties [13]. Techniques of interest providing information on tissue structure and composition are standard histology, transmission electron microscopy, immunohistochemistry, reverse transcription-polymerase chain reaction RT-PCR, Western Blot, and ELISA. Another important outcome is the assessment of tendon mechanical properties. These measurements provide information on viscoelastic properties and mechanical resistance which are altered during tendinosis [13].

To date, management of Achilles tendinopathy remains challenging. In the acute phase, initial rest is important as well as mechanical load control, modification of training regimes, and specific exercises. The benefits of NSAIDs are reliving pain in the acute phase but their use in chronic tendinopathy is questionable. Exercise-based rehabilitation programs are essential components of the treatment, and eccentric muscle loading has become the dominant conservative intervention strategy for Achilles tendinopathy [18]. Multiple studies and systematic reviews have found that eccentric exercises are beneficial in the early stage of AT [19,20] with parallel morphological changes observed in ultrasound: normalized tendon structure, decreased thickness, and reduced neovascularization [21,22]. However, only 60% of athletic and non-athletic patients benefited from eccentric training [23,24], and adherence to exercise programs is not as high in real life as it is in clinical trials. Thus, in addition to eccentric exercises, various nonoperative therapies have been proposed to improve symptoms, such as ultrasound, extracorporeal shockwave therapy (ESWT) or laser therapy [25] or injections (corticosteroids, platelet-rich plasma (PRP)). Therapeutic ultrasound is a commonly prescribed program of physical therapy and in animal studies, ultrasound could stimulate collagen synthesis in tendon fibroblasts. However, due to a lack of high-quality data on the effect of ultrasound on Achilles tendinopathy, the evidence remains insufficient to support its clinical use [3]. Another physical therapy is extracorporeal shockwave therapy (ESWT). Two modalities of ESWT have been studies in AT: focus (maximal energy delivered into a focused point at a predetermined tissue depth) or radial shock waves (energy dissipated over a large area). The two modalities have not been compared in Achilles tendinopathy [26]. ESWT is a safe and well-tolerated treatment modality but there remains a need for further studies on the effectiveness of shock waves due to the complexity of the biological response to shock waves, the high diversity of application methodologies, and the lack of objective measurements in published studies [26]. Peritendinous corticosteroid injections provide short-term relief of pain but their use is limited in Achilles tendinopathy due to reported cases of rupture after injection. More recently, injections of platelet-rich plasma (PRP) at the site of tendon injury has been used in AT because it is thought to facilitate healing through the growth factors it contains [27]. Finally, surgical treatment may be considered after at least six months of nonoperative management. Surgical options include longitudinal tenotomies, release of peritendinous adhesions, excision of the paratenon, and debridement of degenerative tissue [28].

Despite these multiple therapeutic options, there is no established therapeutic strategy in AT due to a lack of evidence or controversies about their efficacy. The objective of the review was to collect and comment on published data on percutaneous procedures used in Achilles tendinopathy, focusing on their structural and molecular effects on the tendon tissue. We will first discuss data about local injections (platelet-rich plasma, corticosteroids, anti-VEGF, sclerosing agents or stem cells) and secondly about the percutaneous needle techniques without injections (tenotomy/dry needling). In each section, we add experimental data to the results of clinical trials when available.

## 2. Ultrasound-Guided Injections

### 2.1. Platelet-Rich Plasma

Among injected treatments, injections of platelet-rich plasma are the most studied. The hypothesis that PRP can promote tissue healing is based on the high content of growth factors in the granula of the platelets especially transforming growth factor (TGF-β), platelet-derived growth factor, insulin-like growth factor 1, vascular endothelial growth factor (VEGF), and fibroblast growth factor which are known to regulate angiogenesis and tissue regeneration [29]. In vitro, PRP was able to induce tenocyte proliferation [27,30,31] and to increase cell viability [32]. It also induced the expression of collagen I [30,31] and MMP1 and *MMP3* by tenocytes [31,32]. In addition, it could promote angiogenesis at the early stage of the healing process by increasing *VEGF* expression [31]. Finally, PRP may have anti-inflammatory effects since it was able to downregulate the production of IL-6 and IL-8 by IL1β-exposed tenocytes [33] to decrease their level of expression of *cyclooxygenase 1 (COX-1)*, *COX-2*, and *PGE2* [34].

Based on these in vitro results, intra-tendinous injections of PRP were used in animal models of Achilles tendinopathy, especially the collagenase-induced model. When PRP was injected early after the collagenase-induced injury (3 days or 7 days), a reduction of tendinopathic lesions was observed. While collagenase injection induced an increase in tendon thickness, fiber disorganization, and neovascularization, PRP injection significantly reduced the structural damages compared to the control group [35,36,37] (Table 1). It also improved the fibrillary structure assessed by transmission electron microscopy (TEM) [36,37]. A significant increase in expression levels of type I collagen was observed after PRP injection compared to saline injection [36,37,38,39] but its effects on MMPs remains uncertain [36,38]. As in vitro, PRP showed an anti-inflammatory potential in animal studies since it was able to increase IL-10 expression [36,37] and decrease IL-6 expression compared to saline injection [36,38]. It was also able to downregulate *PGE2* production and *COX-1* and *COX-2* expression after an Achilles injury [34] (Table 1). In contrast, PRP injections did not have any impact on local concentrations of IL-1α, IL-1β, IL-18, G-CSF, GM-CSF, M-CSF, MIP-1α, RANTES, and TNF-α [40]. Finally, PRP could also influence angiogenesis by increasing VEGF and VEGF-Receptor expression [37]. During the course of tendinopathy, the tendon has altered elastic properties and less tensile strength. The impact of PRP on mechanical properties is not well established in animal studies because the results are contradictory across studies: the parameters were either improved [37,39,41] or unchanged [36,42] (Table 1).

Among growth factors present in PRP, the contribution of each in the observed effects is not defined. One study evaluated the role of HGF (hepatocyte growth factor). It was considered by the authors to be the one responsible for the anti-inflammatory effects of PRP [34]. Both in vitro and in vivo, the expression of *COX-1* and *COX-2* and the production of PGE2 were suppressed by adjunction of PRP, and this suppressive effect was abolished by anti-HGF antibodies, supporting this hypothesis. However, further studies are needed to determine the exact role of the different growth factors present in PRP.

Together, these data suggest that injections of PRP at the site of tendinopathy could stimulate the production of collagen by tenocytes to accelerate tendon healing and could inhibit inflammation, limiting the release of pro-inflammatory IL-6 and increasing the production of IL-10. Although laboratory data and open-label studies or retrospective series showed promising results of intra-tendinous injections of PRP in chronic AT, randomized controlled trials failed to demonstrate a superior clinical efficacy compared to saline injection [43,44]. There was no difference in terms of ultrasound lesions (tendon thickness and Doppler activity). Several points may explain the discrepancy between animal and human research. Firstly, the collagenase-induced model represents an acute inflammatory model and it does not really mimic chronic tendinopathy. Secondly, in the animal experiments, injections are performed quickly after the onset of tendinopathy while in clinical trials, the patients suffer from chronic lesions. Thirdly, the randomized controlled trials included a relatively small number of patients (about 20 patients per group of treatment), and larger trials could have different results.

### 2.2. Corticosteroids

Peri-tendinous corticosteroids (CS) are commonly used because they are beneficial in the short term to reduce the pain of patients suffering from tendinopathy [45]. In AT, Wetke et al. recommend the use of CS injections in patients if rehabilitation alone does not lead to improvement [46]. However, ruptures of the Achilles tendon have been reported, questioning their use in AT [47]. In addition, the mechanisms by which corticosteroids induce pain relief in tendinopathies are not clearly identified.

Corticosteroids are anti-inflammatory drugs, and this effect was confirmed when dexamethasone was applied to tenocytes in vitro. Dexamethasone was able to reduce the expression levels of pro-inflammatory cytokines *IL1-α*, *IL-1β*, and *IL-6* by tendon cells [48] (Table 2). To evaluate mechanisms leading to pain relief, the regulation of substance P (SP) production by CS has been investigated [48]. SP is a neurotransmitter with activity in nociception and inflammatory phenomena, and it has been previously shown that the pain in rotator cuff tendinopathy was correlated with the local tissue level of substance P [49]. In addition, SP can be endogenously produced by tendon fibroblasts when subjected to mechanical loading. Exposure to dexamethasone modulated SP production by human tenocytes stimulated by IL-1β or subjected to cyclic loading [48] suggests that this mechanism would potentially be involved in the relief of pain after CS injections. In vitro, corticosteroids could also influence production of MMP by tenocytes: treatment with triamcinolone acetonide reduced the expression of *MMP2*, *MMP8*, *MMP9*, and *MMP13* by human tenocytes [50] (Table 2).

Few studies are available studying the impact of CS on the mechanical resistance of tendons [42,51] (Table 2). In the two studies, corticosteroid injected into the Achilles tendon did not alter the biochemical resistance of tendons. However, it is difficult to extrapolate these results to humans considering the site of injection (into the tendon and not around it) and the use of non-pathologic tendons for one of the two studies [51]. In another study with CS injected around normal Achilles tendons, there was a transient increase in the number of apoptotic cells in the surface layers of the tendons and a decrease in biomechanical strength one week after the injection, but this phenomenon was self-resolving with a return to a normal strength after three weeks [52]. These data suggest that a rest period should be recommended to the patient if a corticosteroid peritendinous injection is performed.

Corticosteroids have also been used in high-volume injections (HVI). A high-volume injection consists of a large volume of saline, steroid, and local anesthetic under ultrasound guidance into the interface between the midportion of the Achilles tendon and Kager’s fat pad. HVI with corticosteroid showed better short-term effects than HVI without steroid [53], and HVI with steroid seemed more effective than PRP and the placebo in the short term [54].

Despite their common use in Achilles tendinopathy, PRP injections lack strong evidence of efficacy, and CS injections only have a short-term effect on pain, so there is a need for new therapies. More recently, therapeutics based on injections of stem cells have been proposed to treat tendon pathologies.

### 2.3. Stem Cells and Autologous Tenocyte Injections

Injections of stem cells belong to regenerative medicine, which is based on the development of therapies to regenerate or replace injured, diseased, or defective cells, tissues, or organs. Mesenchymal stem cells (MSCs) are non-hematopoietic stromal cells that can differentiate into bone, cartilage, muscle, ligament, tendon, and adipose tissue. It was hypothesized that MSCs could have the ability to create an optimal environment to support tendon regeneration through extracellular matrix synthesis, production of growth factors, stimulation of vessels formation, and replacement of damaged cells [55,56]. Stem cells can be obtained from a variety of sites: they can be derived from bone marrow or adipose tissue. Adipose tissue is an attractive cell source for stem cells because it is ubiquitous and easily obtained in large quantities with little donor site morbidity and discomfort. Stem cells can also be extracted from tendons. Tendon-derived stem cells have the advantage of being lineage-specific cells.

A few studies on their use in animal models of tendinopathy are now available (Table 3). Several types of stem cells have been used: bone marrow-derived MSC [41,57], adipose-derived stem cells (ASCs) [58], autologous tenocytes [59], and tendon-derived stem cells (TDSCs) [39,60]. After their injection, MSCs derived from bone marrow and autologous tenocytes were always present at the site of collagenase injury 6 or 8 weeks later and incorporated into the ECM [57,59]. Most of the time, injections of stem cells led to a reduction of histologic lesions induced by collagenase injection considering the organization of the collagen fibers, cellularity, rounding of the nuclei, and neovascularization [57,58,59,60] (Table 3). The impact of stem cells on mechanical property recovery remains uncertain because the improvement in mechanical strength was not constant (Table 3).

Exosomes derived from various cells seem to be an interesting research axis for regenerative medicine. Exosomes are nanosized [70–150 nm] membrane-bound extracellular vesicles, involved in cell-to-cell communication and also in the development of tissue injury repair. They seem to have the same capacity as tendon stem cells to induce tendon healing [60].

Finally, some authors proposed injecting stem cells combined with PRP [39,41]. It was hypothesized that PRP may have the ability to promote tenocyte differentiation of stem cells and could potentiate their propensity to regenerate the tendon. However, the little data available do not allow us to conclude whether there is a synergistic effect of the two treatments because the two studies available showed opposite results. The combined injection of TDSCs + PRP reduced histological lesions of tendinopathy and improved tensile strength compared to PRP, TDSCs or the control group [39] but the combined injection of bone marrow-derived stem cells + PRP did not improve histologic scores (morphology and density of tenocytes, neovascularization, inflammatory cell infiltrate, linearity and undulation of collagen fibers, and thickness) or the mechanical strength of Achilles tendons [41].

With regard to human tendinopathy, no studies using stem cells to treat Achilles tendinopathies have been published to date. A study is registered on Clinicaltrials.gov and is still recruiting. The ASCAT study is a phase IIA, proof-of-concept study on 10 patients [61]. The patients will received autologous bone marrow-derived stem cells at the area of greatest degeneration of their Achilles tendon and will be followed for six months. Two preliminary studies have been published using stem cells injected under ultrasound at other sites of tendinopathy [62,63]. The injection of skin-derived tenocyte-like cells within the patellar tendon led to a greater improvement in pain and function compared to the injection of plasma as a control [62]. In chronic lateral epicondylitis, the injection of AT-MSCs mixed with fibrin glue induced a decrease in VAS scores from 66.8 ± 14.5 mm to 14.8 ± 13.1 mm and a reduction in hypoechoic lesions observed by ultrasonography [63]. However, there was no control group in this study.

Data available on stem cell therapies in tendinopathies are promising, and studies are needed in the field of Achilles tendinopathies to determine whether the supply of stem cells could contribute to tendon healing and improve patients’ symptoms.

### 2.4. Sclerosing and Anti-Angiogenic Agents (AA)

Tendon neo-angiogenesis appears to be an important step in the healing process through the supply of nutrients and oxygen, the transport of healing factors, and the clearance of cellular debris. However, the persistence of this hypervascularization could be deleterious. Indeed, an imbalance between pro- and anti-angiogenic factors could promote abnormal angiogenesis consisting of defective vessels leading to hypoxia and disorganization of the extracellular matrix [64]. VEGF, the main promoter of this neovascularization, is also able to stimulate the production of MMPs and the inhibition of TIMPs, facilitating the degradation of the ECM [65]. In clinical studies, the beneficial or deleterious role of neovascularization is discussed. Indeed, its correlation with clinical course is not clearly established due to the inconsistency of the studies.

Bevacizumab, an anti-VEGF antibody, has been tested in an animal model of Achilles tendinopathy. The antibody was injected at the site of the collagenase injury [66]. On day 6, ultrasound showed a reduction in the tendon thickness of Achilles tendons in the bevacizumab group compared to saline and less disorganized collagen fibers in the treated group, but on day 13, there was no longer any difference between the two groups. However, the combination with PRP seemed to be more interesting and led to a better healing response than PRP alone [67]. The authors hypothesized that anti-angiogenic agents injected early after the development of tendinopathy might act positively by preventing deleterious proteolytic enzymes and prostaglandins caused by neoangiogenesis. However, later in the healing process, these neovessels provide active growth factors that stimulate healing. Therefore, the concomitant PRP injection would compensate for neo-vessel destruction by locally providing important concentrations of active growth factors able to promote stem cell recruitment and collagen production.

Sclerosing agents have also been proposed to treat Achilles tendinopathy when neovascularization was present. The injections are performed with the aid of grey-scale ultrasonography and color Doppler, to inject the target vessels localized around the tendon. The treatment is injected in fractions until the vessels are no longer visible (any remaining circulation in the neovessels around and inside the Achilles tendon). In clinical studies, their effect on patients’ symptoms is variable, and the few studies available have found conflicting results [68,69,70]. Considering sclerosing agents, no animal studies are currently available.

Although angiogenesis and regulation of vascularization are important components of the pathogenesis of AT, only a few data are available on the contribution of treatments that inhibit angiogenesis in the treatment of AT. Further studies are needed to determine whether tendinous neovascularization should be a therapeutic target.

### 2.5. MMP Inhibitors (Aprotinin)

MMPs and their inhibitors (TIMPs) are crucial to ECM remodeling, and a balance exists between them in normal tendons. Alteration of MMP and TIMP expressions from basal levels leads to alteration of tendon homoeostasis [71]. During tendinopathy, tendons have an increased level of matrix remodeling, leading to a mechanically less stable tendon which is more susceptible to damage [72]. Changes in MMP and TIMP expression levels have been observed [6,71]. On this basis, aprotinin, a MMP inhibitor, has been tested to treat Achilles tendinopathy. Despite open-label studies suggesting an effect on symptoms of AT [73,74], peri-tendinous injections of aprotinin were not shown to offer any statistically significant benefit over the placebo in a double-blind placebo controlled trial [75]. No data are currently available on their local effects in animal models.

To conclude, the different injectable therapies have been developed on the basis of known pathogenesis data for tendinopathies. In animal studies, PRP has the ability to improve tendon structure and resistance, but the promising results of animal studies have not been yet replicated in humans. The contribution of therapies targeting neovascularization is not established, and corticosteroids have only a short-term effect. The more recently proposed therapies using stem cells seem interesting, but their use in humans is quite complex, requiring first a sampling phase, then an expansion phase before injection. In Achilles tendinopathy, the first human studies using stem cells are expected.

In the last part of the review, we will review data on percutaneous interventions without injection.

## 3. Ultrasound-Guided Procedures Without Injection

When conservative treatments have failed to improve patients’ symptoms related to AT, surgical treatments can be proposed. The traditional surgical treatment of midportion Achilles tendinosis has for many years consisted of a dorsal approach, with a central longitudinal tenotomy and excision of tendinosis tissue and/or debridement of the paratenon [76]. This method is associated with a relatively long postoperative rehabilitation, and often it takes 3–6 months before the patients are back in full activity. As an alternative to open surgery, percutaneous procedures have been developed. Two types of techniques have been reported: percutaneous soft tissue release targeting the paratendon and percutaneous tenotomy using a scalpel blade or a needle (dry needling).

### 3.1. Percutaneous Soft Tissue Release (PRST)

The purpose of the procedure is to release adhesions around the paratenon which is thickened in chronic tendinopathy (especially between the paratenon and the crural fascia). The ingrowths of sensory and sympathetic nerves from the paratenon with release of nociceptive substances may play an essential role in causing pain. Ultrasound is used to guide the insertion and scraping. The needle is inserted in front of the tendon, and then forward and backward movements are repeatedly performed to release the soft tissues and paratenon around the tendon until the needle is free to move [77]. In murine Achilles tendinopathy, PRST decreased the cellular inflammatory infiltrate present in the paratenon, and a reduction of substance P expression was observed in the dorsal horns of the spinal cord, suggesting a potential impact on pain regulatory mechanisms [77]. In humans, the technique was effective to reduce pain in patients suffering from chronic Achilles tendinopathy in a randomized study compared to mini-open surgery [68].

### 3.2. Percutaneous Tenotomy/Dry Needling

In Achilles tendinopathies when pain persists despite conservative treatment, a surgical option may be considered. The traditional operation involves a longitudinal skin incision, paratenon incision and stripping, multiple longitudinal tenotomies, and, if a definite area of degeneration is found, its excision. The aim of this intervention is to promote tendon healing induced by a modulation of the tendon cell–matrix environment. In an effort to shorten recovery and reduce morbidity, less invasive approaches have been studied. Percutaneous ultrasound-guided longitudinal tenotomies using a scalpel blade have been proposed [78], and case series have reported interesting results on pain and function [78,79]. Tenotomies using a needle instead of a scalpel blade have also been performed.

The hypothesis is that dry needling could stimulate a healing response. Repeated passages of the needle would produce physical trauma to the tendon which induces bleeding and in turn releases growth factors that stimulate a healing response [80]. Another hypothesis is that passages of the needle could change a chronic tendon injury into an acute inflammatory state leading to the formation of granulation tissue. This granulation tissue would strengthen the tendon. The needle passes are made through the abnormal region of the tendon with real-time ultrasound imaging for continual guidance.

After needling, transcription factors involved in proliferation and differentiation of mesenchymal cells (*Egr1*, *Egr2*, *c-Fos*, and *FosB*) and genes involved in inflammatory responses (*IL-1b*, *IL-1rn*, *Prostaglandin E synthase*, and *iNOS*) or involved in angiogenesis (*VEGF*, *Angiopoietin-1*, and *Angiopoietin-like 1*) were regulated [81]. This suggests that dry needling could stimulate the inflammatory response and the angiogenesis as previously hypothesized. Repeated microtrauma also increased the strength of the tendon callus [81]. In addition, it has been shown in healthy supraspinatus tendons that dry needling can cause a transient increased in vascularity and induce a transient increase in collagen III, TNFα, and IL-1β [82].

Two studies have reported the efficiency of dry needling to improve symptoms of AT [83,84]. No complications were reported after the procedure. Despite its possible role as a regulator of vascularization, no significant changes in neovascularity were observed after dry needling during ultrasound follow-up. Dry needling was also compared to PRP injection in Achilles tendinopathy in a retrospective observational study [85]. At three and six months, no differences in terms of pain and function were observed between the two groups. Dry needling led to a reduction in pain with maximum benefits four months after the procedure when combined with PRP injection [86]. Despite a significant improvement in symptoms, no changes in tendon thickness were observed [84,86].

Although dry needling seems to be an option to trigger tendon healing in the case of tendinopathies, further studies exploring its mechanisms of action are needed. In addition, the few clinical studies available are observational, and new clinical trials with a control group will be necessary to determine its real clinical benefit.

The main data from the review are summarized in Figure 1.

## 4. Conclusions

Although injected treatments or dry needling show promising results in animal models, further studies are needed in humans to determine their position in the therapeutic strategy of Achilles tendinopathy. Similar to PRP, although results seem interesting in animal models, effects on pain and function are not superior to placebos in the randomized clinical trials. This could be explained by the use of collagenase-induced models of tendinopathy which produce an acute tendon lesion and do not reproduce chronic lesions induced by mechanical overload. Although overuse models are longer and more difficult to produce, they would be more interesting in pre-clinical treatment trials. Considering treatments with stem cells, animal studies have demonstrated the ability of these cells to integrate the matrix and adopt a tenocyte phenotype. Stem cell-based therapies could, in the future, be a means of resolving the deficient tendon healing process leading to tendinopathic lesions.

## Figures and Tables

**Figure 1 ijms-21-07000-f001:**
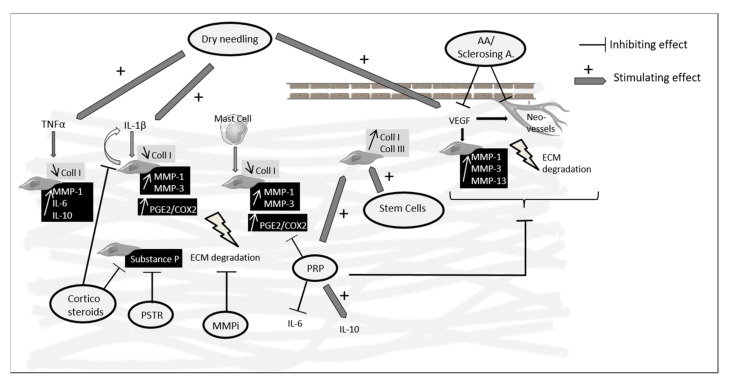
Main targets and effects of percutaneous treatments used in Achilles tendinopathy reported in in vitro and animal studies. PRP and stem cells showed that they were able to increase collagen synthesis and PRP also showed anti-inflammatory effects inhibiting IL-6 and upregulating IL-10. Corticosteroid injections and PSTR led to a decreased expression of substance P which may explain in part their pain-relieving effects. Dry needling was able to stimulate an inflammatory reaction and angiogenesis. MMPi target MMPs which are increased in tendinopathy creating an imbalance in the synthesis/degradation process, but no animal studies are available to know their impact on tendinopathy lesions. AA: anti- angiogenic agents; A.: agents; Coll: collagen; MMP: matrix metalloproteinase; ECM: extracellular matrix; PRP: platelet-rich plasma; PG: prostaglandin; COX: cyclooxygenase; MMPi: MMP inhibitors; PSTR: percutaneous soft tissue release.

**Table 1 ijms-21-07000-t001:** Main results of articles that studied the effects of platelet-rich plasma (PRP) in animal models of Achilles tendinopathy. All studies were controlled, and significant effects compared to the control group are reported. D: day; W: week; C: collagenase; I: injury; Coll: collagen; COX: cyclooxygenase; PGE2: prostaglandin E2; MMP: matrix metalloproteinase. IL: interleukin, TNFα: Tumor necrosing factor alpha; G-CSF: Granulocyte-colony stimulating factor; GM-CSF: Granulocyte Macrophage-colony stimulating factor; M-CSF: Macrophage-colony stimulating factor; MIP-1α: Macrophage inflammatory protein-1 alpha; Lr: leucocyte-rich; Lp: leucocyte-poor; ↗: increase; ↘: decrease.

Author, Year	Type of Experiment			Main Results		
	Animal Model	Time of Treatment	Time of Analysis	Histology	Gene and Protein	Biomechanical Testing
Zhang, 2013 [31]	Wound of 1 mm Mouse	D0	D0, D1, D3, D5 and D12 after I	-	- ↘ PGE2 production after injury - ↘ *COX-1* and *COX-2* expression	-
Dallaudière, 2013 [32]	Collagenase Rat	D3	D6, D13, D18 and D25 after C	- Reduction of tendon thickening and fibrillary disorganization as well as neovascularization with PRP	-	-
Li, 2020 [33]	Collagenase Rabbit	D7 or W4	6 weeks after C	- Significant reduction of histological lesions (PRP injected at D7)	- ↗ expression of *Coll1* (D7) and ↗ expression of *Coll3*, *MMP1* and *MMP3* (W4) - ↗ level of IL-10 and ↘ level of IL-6 - No difference in IL-1β and TNFα concentrations	- No improvement in failure load and stiffness
Jiang, 2020 [34]	Collagenase Rabbit	D7 (Lp-PRP, Lr-PRP and saline)	3 and 6 weeks after C	- Significant reduction in histological lesions in the 2 PRP groups compared with saline but better in Lr-PRP	- ↗ expression of *Coll1* earlier in Lr-PRP group - ↗ then ↘ *VEGF* and *VEGF-R* expression (Lr-PRP group) - Transient ↗ IL-10 expression (Lr-PRP group)	- Failure load, stiffness, and tensile stress in the Lr-PRP group higher than those in the saline group
Yan, 2017 [35]	Collagenase Rat	W4 (Lp-PRP, Lr-PRP and saline)	8 weeks after C	- Reduction of tendinopathic lesions compared to saline (Lp-PRP)	- ↗ expression of *Coll1* compared with the saline group (Lp-PRP) - No impact on IL-1β and TNFα concentrations but ↘ of IL-6 level compared to saline (Lp-PRP) - ↘ *MMP1* and *MMP3* expression. No effect on *MMP9*	-
Chen, 2014 [36]	Collagenase Rat	W4	8 or 12 weeks after C	- Significant improvement of histological parameters of tendon quality (fiber organization, nuclear rounding, and inflammation)	- ↗ expression of *Coll1, scleraxis* and *Tenascin C*	- Maximum load to failure and stiffness significantly superior to the control group by week 8
Dallaudière, 2015 [37]	Collagenase Rat	D3	D7, D13, D18 and D25 after C	-	No effect on local concentrations of IL-1α, IL-1β, IL-18, G-CSF, GM-CSF, M-CSF, MIP-1α, RANTES, TNFα	-
Fedato, 2019 [38]	Collagenase Rat	D5	4 weeks after PRP	- No significant effect on histological lesions compared to control group	-	- Better results for maximum deformation and elastic modulus. Ultimate tensile strength not improved.
Solchaga, 2014 [39]	Collagenase Rat	D7	D14 and D28 after C	- No differences between groups in the extent and character of the repair	-	- No improvement in mechanical properties (maximum load, ultimate tensile stress, stiffness)

**Table 2 ijms-21-07000-t002:** Main results of articles that studied the effects of corticosteroids on Achilles tendinopathy: in vitro studies with tenocytes and animal models. All studies were controlled, and significant effects compared to the control group are reported. D: day; W: week; C: collagenase; I: injection; Coll: collagen; MMP: matrix metalloproteinase; TIMP: Tissue inhibitor of metalloproteinase; TCA: triamcinolone acetonide; PSL: prednisolone. IL: interleukin; SP: substance P; ↗: increase; ↘: decrease.

Author, Year	Design			Main Results		
	Type	Cells/Model	Steroid	Histology	Gene and Protein	Biomechanical Testing
Mousavizadeh, 2015 [45]	In vitro	Human tenocytes	Dexamethasone	-	- ↘ expression of *IL-1**α*, *IL-1**β*, and *IL-6* - ↘ expression level of *TAC1* (gene encoding for substance P (SP)) but no effect on the expression of *NK1R*, its receptor. Reduced secretion of SP - ↘ induction of SP by IL-1β and by mechanical loading	-
Tempfer, 2009 [47]	In vitro	Human tenocytes	Triamcinolone acetonide	-	- ↘ Expression and secretion of Coll1 - ↘ proliferation rate - ↘ expression of *MMP2*, *MMP8*, *MMP9*, and *MMP13* but ↗ expression of *TIMP1* - Increased expression of *SOX9*	-
Solchaga, 2014 [39]	In vivo	Collagenase Rat	Intra-tendinous triamcinolone acetonide at D7 Analysis at D14 and D28 after C	- ↘ tendon thickness at insertion site and midsubstance - ↘ cell proliferation and ↘ inflammation relative to saline treatment	-	- No effect on mechanical properties compared to saline (maximum load, ultimate tensile stress, stiffness, ramping modulus)
Dinhane, 2019 [48]	In vivo	Normal Achilles tendons Rabbit	Intra-tendinous injection of betamethasone Analysis 48 h after I	-	- ↘ *MMP2* expression compared with the control group - No difference in IL1 and IL6 levels in the tendon tissues	- No effect on mechanical resistance (maximum deformation, maximum force, energy at maximum force, elasticity modulus, and tension at maximum force)
Muto, 2014 [49]	In vivo	Normal Achilles tendons Rat	Triamcinolone acetonide (TCA) or prednisolone (PSL) around the tendon. Analysis at W1 and W3 after I	- Collagen fiber bundles irregularly aligned at W1 but at W3, these changes had ↘. - At W1, increased number of apoptotic cells in the surface layers of the tendons but no significant difference after 3 weeks	- at 1 week, ↗ MMP3 in the surface layers of tendons	- Reduction of maximum failure load 1 week after the injection with a return to normal after 3 weeks

**Table 3 ijms-21-07000-t003:** Main results of articles that studied the effects of stem cell therapies in animal models of Achilles tendinopathy. All studies were controlled, and significant effects compared to the control group are reported. D: day; W: week; C: collagenase; I: injury; Coll: collagen; MSC: mesenchymal stem cells; TDSC: tendon-derived stem cells; PRP: platelet-rich plasma; ttt: treatment. IL: interleukin; SP: substance P; ↗: increase; ↘: decrease.

Author, Year	Design			Main Results		
	Animal Model	Treatment	Time of Analysis	Histology	Gene and Protein	Biomechanical Testing
Machova Urdzikova, 2014 [52]	Collagenase Rat	Bone marrow-derived hMSC injected at D3	W2, W4 and W6 after C	- Better organization of the collagen fibers and ↗ neovascularization in hMSC-treated rats - MSC always present at the site of injection 6 weeks later	- ↗ amounts of Coll1 and Coll3 - No difference in quantities of aggregan and versican in the ECM	- No difference between groups (testing of stiffness and load to failure)
Oshita, 2016 [53]	Collagenase Rat	Adipose-derived stem cells (ASCs) injected at D7	W4 and W12 after ttt	- ↘ degree of tendon degeneration (↘ disrupted collagen fibers, ↘ cellularity, and less ground substance deposition between collagen fibers)	- ↘ *Coll3*/*Coll1* ratio in the ASC group	-
Chen, 2011 [54]	Collagenase Rat	Autologous tenocytes injected at W4	4 and 8 weeks after ttt	- Histologic scores significantly better at W8 (fiber structure, rounding of the nuclei, inflammatory cells, and neovascularization) - Tenocytes incorporated into the ECM and distributed longitudinally and parallel to the fiber orientation in a typical spindle pattern	- At W8, ↗ synthesis of Coll1 compared to the control group	- Improvement of the ultimate failure load
Chen, 2014 [36]	Collagenase Rat	Tendon-derived stem cells injected at W4	8 or 12 weeks after C	- Significant improvement of histological parameters (fiber arrangement and structure, nuclear rounding, and inflammation) after combined injection of TDSC and PRP. - No effect of TDSC alone.	- No effect of TDSC on tenocyte-related gene expression (*Coll1*, *Scleraxis*, *Tenascin C*) and or non-tenocyte gene expression (*Runx2*, *SOX9* and *PPARγ*)	- Maximum load to failure and stiffness significantly superior to the control group when TDSCs were injected with PRP but no effect of TDSC alone on mechanical parameters.
Fedato, 2019 [38]	Collagenase Rat	Bone marrow-derived stem cells injected at D5	4 weeks after ttt	- No significant improvement of histological lesions	-	- Significantly better results for elastic modulus in the stem cell group - Ultimate tensile strength and maximum deformation not improved.
Wang, 2019 [55]	Collagenase Rat	Tendon-derived stem cells injected at D7	5 weeks after collagenase	- Reduction of histological lesions	-	- ↗ Maximum loading and ultimate stress in the TDSC group compared with the control group

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
