# Peer review of "Molecular and Structural Effects of Percutaneous Interventions in Chronic Achilles Tendinopathy"

_ijms, 2020, doi:10.3390/ijms21197000_

Round 1

Reviewer 1 Report

This is an interest review of a trend topic as ultra sound  guided treatments in achilles tendinopathy. 

Some issues could be improved in order to achieve a better scientific quality. 

Introduction section needs to be improved in order to achieve a deeper knowledge. 

Please could you provide at the end of the introduction the main purpose? 

Could you provide a methods section in order to explain how the different techniques were selected? Could you provide how the articles research was searched (mesh, booleans...) In different clinical database?

Could you provide a table with the different quality level of the selected manuscripts in rererences in order to know bias, randomization, selection etc etc. (Caspe, Pedro and Van Tulder) 

Could you provide a discussion section? 

Kind Regards

Author Response

Nantes,

August 15, 2020

First, I want to thank the reviewer for the relevant comments. The point-by-point responses are given below.

Response to reviewer#1:

This is an interest review of a trend topic as ultrasound guided treatments in achilles tendinopathy. Some issues could be improved in order to achieve a better scientific quality.

Point 1: Introduction section needs to be improved in order to achieve a deeper knowledge.

Several points have been added to complete the introduction section especially about therapeutic management of Achilles tendinopathy. We completed the paragraph line 88 : “Multiple studies and systematic reviews have found that eccentric exercises are beneficial in the early stage of AT [16,17] with parallel morphological changes observed on ultrasound : normalized tendon structure, decreased thickness and reduced neovascularization [18,19].

Line 92, we added: “Thus, in addition to eccentric exercises, various nonoperative therapies have been proposed to improve symptoms, such as ultrasound, extracorporeal shockwave therapy (ESWT) or laser therapy [22] or injections (corticosteroids, platelet-rich plasma (PRP)). Therapeutic ultrasound is a common prescribed program of physical therapy and in animal studies, ultrasound could stimulate collagen synthesis in tendon fibroblasts. However, due to a lack of high quality data on the effect of ultrasound on Achilles tendinopathy, the evidence remains insufficient to support its clinical use [3]. Another physical therapy is extracorporeal shockwave therapy (ESWT). Two modalities of ESWT have been studies in AT: focus (maximal energy delivered into a focused point at a predetermined tissue depth) or radial shock waves (energy dissipated over a large area). The two modalities have not been compared in Achilles tendinopathy [23]. ESWT is a safe and well-tolerated treatment modality but it remains a need for further studies on the effectiveness of shock waves due to the complexity of the biological response to shock waves, the high diversity of application methodologies, and the lack of objective measurements in published studies [23].”

We also add ref [24] about PRP and ref [25] about surgical treatments.

Point 2: Please could you provide at the end of the introduction the main purpose?

The objective of the review has been reworded. Line 115, we added:” The objective of the review was to collect and comment published data on percutaneous procedures used in Achilles tendinopathy, focusing on their structural and molecular effects on the tendon tissue. We will first discuss data about local injections (platelet-rich plasma, corticosteroids, anti-VEGF, sclerosing agents or stem cells) and secondly about the percutaneous needle techniques without injections (tenotomy/dry needling). In each section, we will put in parallel experimental data with the results of clinical trials when available.”

Point 3: Could you provide a methods section in order to explain how the different techniques were selected? Could you provide how the articles research was searched (mesh, booleans...) In different clinical database?

When we were asked to contribute to the special issue, we proposed a narrative review rather than a systematic review aimed at answering a specific question. In this review, we have proposed to synthesize data on the effects of the local treatments studied in Achilles tendinopathy based on the known elements of pathogenesis which may help to identify future research directions in the field.

To write the review, we did not use the methodology of a systematic review.

Point 4: Could you provide a table with the different quality level of the selected manuscripts in references in order to know bias, randomization, selection etc etc. (Caspe, Pedro and Van Tulder)

We have provided tables summarizing the main findings of the articles cited and their methodology about plasma rich plasma (Table 1, page 5), corticosteroids (Table 2, page 9) and stem cells therapies (Table 3, page 11).

Point 5: Could you provide a discussion section?

We did not include a separate discussion but included point of discussion in the sections because the article is not a systemic review or an original article but a narrative review.

Reviewer 2 Report

The title is somewhat misleading. Some of the interventions discussed are not ultrasound guided. 

Line 86: please be more specific or define "ultrasound". Therapy or imaging?

Line 96-100: include reference. In fact, please review the manuscript for several places were references are needed following statements. 

The review reads more like a list of papers/topics rather than a synthesis and integration of previous literature. Several sections, e.g. 234-237, 253-256, most of 258-265 are note needed. 

Please provide a more detailed description of the figure and how the figure was developed.

How is this review difference from other similar reviews?

Author Response

Nantes,

August 15, 2020

First, I want to thank the reviewer for the relevant comments. The point-by-point responses are given below.

Response to reviewer#2:

Point 1: The title is somewhat misleading. Some of the interventions discussed are not ultrasound guided.

The Title has been modified. The new title is “Percutaneous interventions to treat chronic Achilles tendinopathy: what are their molecular and structural effects?”

Point 2: Line 86: please be more specific or define "ultrasound". Therapy or imaging?

This part was about ultrasound as a therapy. To avoid confusion with imaging, we added the term “therapeutic” in this part. The changes appear line 95:  Therapeutic ultrasound is a common prescribed program of physical therapy and in animal studies, ultrasound could stimulate collagen synthesis in tendon fibroblasts.”

Point 3: Line 96-100: include reference. In fact, please review the manuscript for several places were references are needed following statements.

Several references have been added to support some statements:

- Line 94: [22] Rhim et al. (2020) Comparative Efficacy and Tolerability of Nonsurgical Therapies for the Treatment of Midportion Achilles Tendinopathy: A Systematic Review With Network Meta-analysis.

- Line 102 and 105: [23] Stania et al. (2019) Extracorporeal Shock Wave Therapy for Achilles Tendinopathy

- Line 109: [24] Wang et al. (2012) Proliferation and differentiation of human tenocytes in response to platelet rich plasma: an in vitro and in vivo study

- Line 112: [25] Tallon et al. (2001) Outcome of surgery for chronic Achilles tendinopathy. A critical review

- Line 175: [41] Madhi et al. (2020) The use of PRP in treatment of Achilles Tendinopathy: A systematic review of literature. Study design: Systematic review of literature

- Line 200: [46] Gotoh et al (1998) Increased substance P in subacromial bursa and shoulder pain in rotator cuff diseases

- Line 237: [50] Ahmad et al. (2012) Exploring the Application of Stem Cells in Tendon Repair and Regeneration

- Line 237: [51] Ruzzini et al. (2012) Stem cells and tendinopathy: state of the art from the basic science to clinic application

- Line 301: [59] Hall et al. (2010) Regulation of tumor angiogenesis by the local environment

- Line 340: [67] Del Buono et al. (2013) Metalloproteases and tendinopathy.

- Line 366: [69] Maffulli et al. (2020) Achilles tendinopathy

- Line 402: [73] Stoychev et al. (2020) Dry Needling as a Treatment Modality for Tendinopathy: a Narrative Review

Point 4: The review reads more like a list of papers/topics rather than a synthesis and integration of previous literature. Several sections, e.g. 234-237, 253-256, most of 258-265 are note needed.

- The paragraph between line 253 and 256 has been removed because these points are detailed above.

- The paragraph about stem cells therapies in human trials (between line 258 and 277) have been modified to be more synthetic. It is now line 282: “Two preliminary studies have been published using stem cells injected under ultrasound in other sites of tendinopathy [57,58]. The injection of skin-derived tenocyte-like cells within the patellar tendon led to a greater improvement in pain and function compared to the injection of plasma as control [57]. In chronic lateral epicondylitis, the injection of AT-MSCs mixed with fibrin glue induced a decrease in VAS scores from 66.8 ± 14.5 mm to 14.8 ± 13.1 mm and a reduction in hypoechoic lesions observed by ultrasonography [58]. However, there was no control group in this study.”

Point 5: Please provide a more detailed description of the figure and how the figure was developed.

The figure was developed to schematize the different targets and effects of the treatments considered. The legend of the figure has been completed for a better understanding.

The new title and legend are line 430: “Figure 1. Main targets and effects of percutaneous treatments used in Achilles tendinopathy reported in in vitro and animal studies. PRP and stem cells showed that they were able to increase collagen synthesis and PRP also showed anti-inflammatory effects inhibiting IL-6 and upregulating IL-10. Corticosteroid injections and PSTR led to a decreased expression of substance P which may explain in part their pain-relieving effects. Dry needling was able to stimulate an inflammatory reaction and angiogenesis. MMPi target MMPs which are increased in tendinopathy creating an imbalance in the synthesis/degradation process but no animal studies are available to know their impact on tendinopathy lesions”

Point 6: How is this review difference from other similar reviews?

In this review, we proposed to focus on molecular and structural effects of percutaneous treatments to made a synthesis about positive effects yet demonstrated and discussed then related to results of the clinical trials which are not often as positive as hoped for, thus allowing us to reflect on ways to conduct future research in the field.

Reviewer 3 Report

After carefully reading this manuscript, I must say that, from my point of view, the authors have done research on an important topic: ultrasound-guided interventions in chronic Achilles tendinopathy. This could be interesting to clinicians and researchers in the medical sciences, that frequently work in this area.

It could give them a wider concept about the impact of the overall health of the basis of these therapies.

I have no real problems with the text of this paper, only some suggestions that are mentioned below.

The title should be re-formulated to clarify the study design.

I suggest that background should be improved, with more details about the physical therapies most employed in chronic Achilles tendinopathy.

Were authors considered the addition of a table/figure who support the main findings for each therapy?

Please, authors should be included the doi number in all the references.

Author Response

Nantes,

August 15, 2020

First, I want to thank the reviewer for the relevant comments. The point-by-point responses are given below.

Response to reviewer#3:

After carefully reading this manuscript, I must say that, from my point of view, the authors have done research on an important topic: ultrasound-guided interventions in chronic Achilles tendinopathy. This could be interesting to clinicians and researchers in the medical sciences that frequently work in this area.

It could give them a wider concept about the impact of the overall health of the basis of these therapies. I have no real problems with the text of this paper, only some suggestions that are mentioned below.

I want to thank again the reviewer for the encouraging remarks.  

Point 1: The title should be re-formulated to clarify the study design.

The title has been replaced by “Percutaneous interventions to treat chronic Achilles tendinopathy: what are their molecular and structural effects?”

Point 2: I suggest that background should be improved, with more details about the physical therapies most employed in chronic Achilles tendinopathy.

Several points have been added to complete the introduction section. The paragraph between line 84 and line 113 has been modified and references have been added.

Line 88: “Multiple studies and systematic reviews have found that eccentric exercises are beneficial in the early stage of AT [16,17] with parallel morphological changes observed on ultrasound : normalized tendon structure, decreased thickness and reduced neovascularization [18,19].”

Line 92: “Thus, in addition to eccentric exercises, various nonoperative therapies have been proposed to improve symptoms, such as ultrasound, extracorporeal shockwave therapy (ESWT) or laser therapy [22] or injections (corticosteroids, platelet-rich plasma (PRP)).”

Line 99: “Another physical therapy is extracorporeal shockwave therapy (ESWT). Two modalities of ESWT have been studies in AT: focus (maximal energy delivered into a focused point at a predetermined tissue depth) or radial shock waves (energy dissipated over a large area). The two modalities have not been compared in Achilles tendinopathy [23]. ESWT is a safe and well-tolerated treatment modality but it remains a need for further studies on the effectiveness of shock waves due to the complexity of the biological response to shock waves, the high diversity of application methodologies, and the lack of objective measurements in published studies [23].”

Point 3: Were authors considered the addition of a table/figure who support the main findings for each therapy?

Three tables have been added summarizing the main findings of the studies cited for plasma rich plasma (Table 1, page 5), corticosteroids (Table 2, page 9) and stem cells therapies (Table 3, page 11).

Point 4: Please, authors should be included the doi number in all the references.

The style of the references has been modified to include the doi.

Round 2

Reviewer 2 Report

thank you for addressing my feedback

Reviewer 4 Report

Thank you for having asked me to review this manuscript

Though it has already undergone peer review by other scientists and revision by the authors, I still find in lacking

1. I would avoid the use of a question as a title, and would recommend the use of a simple statement

2. there is no references about the role of MMP and tendinopathy.

3. the role of inflammation in tendinopathy has recently been reviewed https://pubmed.ncbi.nlm.nih.gov/31838495/, as has the role of the immune system

4. validated animal models of tendinopathy are discussed in https://pubmed.ncbi.nlm.nih.gov/20673247/

5. please do not use words such as 'generate' or 'create'. Stick to 'produce' or 'induce'

6. PGE1 has also been used to produce tendinopathy in animals

7. in the ultrasound guided injections, high volume injections should be mentioned and analysed, as they have been tested in randomised trials against PRP (actually being more effective than PRP)

8. line 128: 'yet' should be 'still'

9. MMP inhibitors: it is evident that the authors only read the critiques to the article by Orchard et al. The title of the article is actually 'Successful management of tendinopathy with injections of the MMP-inhibitor aprotinin' , and the text clarifies the issue at hand

10. the authors do not identify the role of ultrasound guided percutaneous tenotomies in the management of Achilles tendinopathy

11. the title is misleading. There is little research on the molecular and structural consequences of many of the treatments described. Indeed, the main outcome measure of these studies, at least in humans, is pain. Histological, gene expression and cell based studies are lacking, probably a consequence of the fact that it would be very difficult to obtain ethics permission to biopsy tendons. In addition, in humans there is little relation between appearance at imaging and clinical symptoms"

Author Response

Here it is the point-by-point replies to the reviewer’s comments. I want to thank the reviewer for the relevant comments.

Replies to the reviewer:

Point 1. I would avoid the use of a question as a title, and would recommend the use of a simple statement.

The title has been modified to replace the question by a simple statement as suggested. The new title is : “Molecular and structural effects of percutaneous interventions in chronic Achilles tendinopathy“.

Point 2. there is no references about the role of MMP and tendinopathy.

To complete references 68 and 69, two references about the role of MMP in tendinopathy were added:

Page 2, line 8:  Riley GP. Gene expression and matrix turnover in overused and damaged tendons. Scand J Med Sci Sports. 2005 Aug;15(4):241-51. doi: 10.1111/j.1600-0838.2005.00456.x. PMID: 15998341

Page 2, line 8 and page 13, line 190: Jones GC, Corps AN, Pennington CJ, Clark IM, Edwards DR, Bradley MM, Hazleman BL, Riley GP. Expression profiling of metalloproteinases and tissue inhibitors of metalloproteinases in normal and degenerate human achilles tendon. Arthritis Rheum. 2006 Mar;54(3):832-42. doi: 10.1002/art.21672. PMID: 16508964

Point 3. the role of inflammation in tendinopathy has recently been reviewed https://pubmed.ncbi.nlm.nih.gov/31838495/, as has the role of the immune system

As suggested, the reference was added page 2 line 7.  

Point 4. validated animal models of tendinopathy are discussed in https://pubmed.ncbi.nlm.nih.gov/20673247/

As suggested, the reference was added page 2 line 28.  

Point 5. please do not use words such as 'generate' or 'create'. Stick to 'produce' or 'induce'

Page 2 line 22: ‘generates’ was replaced by ‘induces’

Page 15 line 301, ‘create’ was replaced by ‘produce’

Point 6. PGE1 has also been used to produce tendinopathy in animals.

This point has been added page 2 line 28 with the accurate reference. 

Point 7. in the ultrasound guided injections, high volume injections should be mentioned and analysed, as they have been tested in randomised trials against PRP (actually being more effective than PRP)

We added a paragraph page 8 line 72 about high volume injections with two additional references: “Corticosteroids have also been used in high-volume injections (HVI). High-volume injection consists of a large volume of saline, steroid, and local anesthetic under ultrasound guidance into the interface between the midportion of the Achilles tendon and Kager’s fat pad. HVI  with corticosteroid showed better short-term effects than HVI without steroid [53] and HVI with steroid seemed more effective than PRP and placebo in the short term [54].  “

Point 8. line 128: 'yet' should be 'still'

As suggested, ‘yet’ has been replaced by ‘still’ on page 12 line 129,

Point 9. MMP inhibitors: it is evident that the authors only read the critiques to the article by Orchard et al. The title of the article is actually 'Successful management of tendinopathy with injections of the MMP-inhibitor aprotinin' , and the text clarifies the issue at hand

In this part on MMP inhibitors, athough only Brown et al. has been cited, we also have read Orchard et al, Maffulli et al., Kearney et al. and Coombes et al. to get an indication of the effectiveness of the treatment. In the study highlighted here, eighty-four percent of patients suffering from AT thought aprotinin injections were helpful, however the study has some limitations (open-label, based on a non-validated questionnaire, no control group). Both the placebo effect and natural improvement of the condition may have contributed to the beneficial effects. The randomized study cited in the manuscript (Brown et al.) also has limitations but it is the only randomized trial available on the subject.

Page 13 line 194, we added this sentence including two additional references: “Despite open-label studies suggesting efficacy on symptoms of AT [73,74], peri-tendinous injections of aprotinin were not shown to offer any statistically significant benefit over placebo in a double blind placebo controlled trial [75].”

Point 10. the authors do not identify the role of ultrasound guided percutaneous tenotomies in the management of Achilles tendinopathy.

We briefly discussed percutaneous tenotomy using a scalpel blade page 14 line 246. Since the review focuses on molecular effects, we have developed more on dry needling in this section because some data on local effects (gene expression regulation, cytokines production) are available whereas this is not the case for scalpel tenotomies.  

Point 11. the title is misleading. There is little research on the molecular and structural consequences of many of the treatments described. Indeed, the main outcome measure of these studies, at least in humans, is pain. Histological, gene expression and cell based studies are lacking, probably a consequence of the fact that it would be very difficult to obtain ethics permission to biopsy tendons. In addition, in humans there is little relation between appearance at imaging and clinical symptoms"

The title has been changed (point 1).

It is true that data on molecular and structural effects of percutaneous treatments used in humans are sometimes weak or absent as reported in the review. As mentioned here, several difficulties explain the lacks in the field of tendinopathy. Further studies will be necessary to better understand the pathogenesis of AT and find the key elements to translate animal data to human pathology.